# Δ*ccr5* Genotype Is Associated with Mild Form of Nephropathia Epidemica

**DOI:** 10.3390/v11070675

**Published:** 2019-07-23

**Authors:** Konstantin Kletenkov, Ekaterina Martynova, Yuriy Davidyuk, Emmanuel Kabwe, Anton Shamsutdinov, Ekaterina Garanina, Venera Shakirova, Ilsiyar Khaertynova, Vladimir Anokhin, Rachael Tarlinton, Albert Rizvanov, Svetlana Khaiboullina, Sergey Morzunov

**Affiliations:** 1Openlab “Gene and Cell Technologies”, Institute of Fundamental Medicine and Biology Kazan Federal University, Kazan 420008, Republic of Tatarstan, Russian; 2Department of Infectious Diseases, Kazan State Medical Academy, Kazan 420012, the Republic of Tatarstan, Russian; 3Department of Pediatric Infectious Diseases, Kazan State Medical University, Kazan 420012, Republic of Tatarstan, Russian; 4School of Veterinary Medicine and Science, University of Nottingham, Loughborough LE12 5RD, UK; 5Department of Microbiology and Immunology, University of Nevada, Reno, NV 89557, USA; 6Department of Pathology, University of Nevada, Reno, NV 89557, USA

**Keywords:** nephropathia epidemica, hemorrhagic fever with renal syndrome, cytokines, matrix metalloprotease, *CCR5*, Δ*32CCR5*

## Abstract

Nephropathia Epidemica (NE), a mild form of hemorrhagic fever with renal syndrome (HFRS) and linked to hantavirus infection, is endemic in the Republic of Tatarstan. Several genetic markers of HFRS severity have been identified previously, including human leukocyte antigen (HLA) complexes and nucleotide polymorphism in the tumor necrosis factor alpha (*TNFα*) gene. Still, our understanding of the genetic markers of NE severity remains incomplete. The frequency of the C–C chemokine receptor type 5 (*CCR5*) gene wild type and gene with 32-base-pair deletion (Δ*32CCR5*) genotypes in 98 NE samples and 592 controls was analyzed using PCR. Along with the serum levels of 94 analytes, a lack of differences in the *CCR5* genotype distribution between NE cases and the general population suggests that the *CCR5* genotype does not affect susceptibility to hantavirus infection. However, in NE cases, significant variation in the serum levels of the host matrix metalloproteases between functional *CCR5* homozygous and Δ*32CCR5* heterozygous patients was detected. Also, the oliguric phase was longer, while thrombocyte counts were lower in functional *CCR5* homozygous as compared to heterozygous NE cases. Our data, for the first time, presents the potential role of the *CCR5* receptor genotype in NE pathogenesis. Our data suggests that NE pathogenesis in functional *CCR5* homozygous and heterozygous NE patients differs, where homozygous cases may have more disintegration of the extracellular matrix and potentially more severe disease.

## 1. Introduction

Nephropathia epidemica (NE), the mild form of hemorrhagic fever with renal syndrome (HFRS) triggered by hantavirus infection, is endemic in the republic of Tatarstan [1]. Kidney failure is commonly associated with the most severe form of NE [2]. The lack of virus replication in the damaged tissues suggests that the disease pathogenesis is most likely the result of the host reaction to virus infection. However, the molecular mechanisms of NE pathogenesis remain largely unknown.

A mononuclear leukocyte infiltration is often found in the kidney tissue of NE cases [3]. The most consistent finding is that these leukocytes include CD8+ T cells and monocytes, suggesting their role in the disease pathogenesis. Mononuclear leukocyte migration is regulated by a group of chemokines including chemokine (C–C motif) ligand 2 (CCL2), chemokine (C–C motif) ligand 3 (CCL3), chemokine (C–C motif) ligand 5 (CCL5), and chemokine (C–C motif) ligand 10 (CXCL10) [4], all of which we have found increased in the serum of NE cases [5].

Chemokine-directed leukocyte migration is regulated by expression of a unique set of receptors, including CCR5 [6]. CCR5 is essential for guiding the migration of activated and effector T cells [7]. Hence, *CCR5* gene polymorphism could affect the severity of virus infection [8,9].

Our data suggests that a CCL5-CCR5 interaction could be involved in the leukocyte migration and tissue accumulation commonly found in NE and HFRS cases [10,11]. This assumption is confirmed by the fact that leukocytes infiltrating NE tissue have a CD8+ and activated T cell phenotype, which is the main target for CCL5-CCR5 driven chemotaxis [10,12]. Although leukocyte accumulation in NE tissues is well-documented, the effect of the *CCR5* genotype on the clinical presentation of NE remains largely unknown.

This study reports that the frequency of functional wild type (wt) *CCR5* homozygous, *CCR5/CCR5* gene with 32-base-pair deletion (Δ*32CCR5*) heterozygous, and Δ*32CCR5* homozygous genotypes is similar between the general population of the Republic of Tatarstan and NE cases, suggesting that susceptibility to hantavirus infection is independent of *CCR5* genotype. Serum levels of matrix metalloproteases (MMPs) and IL9 in functional *CCR5* homozygous NE patients were significantly higher than that in heterozygous NE patients while IL32 levels were decreased (there were too few Δ*32CCR5* patients to analyze). On the other hand, downregulated levels of Platelet Derives Growth Factor BB (PDGFBB), Interferon gamma (IFNγ), IL32 and TNF-related apoptosis-inducing ligand (TRAIL) were associated with severe form of NE. Upregulation of MMP2, creatinine, CSF2, CXCL12, and CCL27 was linked to the mild NE form. Also, the duration of the oliguric phase of the clinical disease was longer, while thrombocyte counts were lower in functional *CCR5* homozygous as compared to heterozygous NE (indicating more severe disease). Our data suggests that the functional *CCR5* homozygous NE may have more pronounced disintegration of the extracellular matrix (ECM) and increased leukocyte trans-endothelial migration as compared to heterozygous cases. Our data, for the first time, presents the potential role of CCR5 receptor genotype in pathogenesis of NE.

## 2. Materials and Methods

### 2.1. Subjects

Serum samples from 98 subjects (80 male and 18 female) admitted to the Agafonov Republican Clinical Hospital for Infectious Disease, Republic of Tatarstan between spring 2015 and fall 2016 were used in the study. Diagnosis of NE was established based on clinical presentation and was serologically confirmed by detection of anti-hantavirus IgG antibodies. Samples were collected following the standard operating protocol for diagnosis of hantavirus infection. Briefly, blood samples were collected into a vacutainer and allowed to coagulate at 37 °C for 30 min. The serum was separated (2000× *g*; 10 min) and stored at −80 °C. Serum from 27 controls (22 male and 5 female) were collected.

Additionally, cheek swabs from the 98 NE patients and 592 donors (423 male and 169 female) were collected and used for DNA extraction.

### 2.2. Ethics Statement

The Ethics Committee of the Kazan Federal University approved this study, and informed consent was obtained from each NE patient and controls according to the guidelines approved under this protocol (article 20, Federal Law “Protection of Health Right of Citizens of Russian Federation” N323-FZ, 21 November 2011).

### 2.3. Hantavirus ELISA

The Hantagnost diagnostic ELISA kit (Institute of Poliomyelitis and Viral Encephalitis, Moscow, Russia) was used to determine the hantavirus-specific antibody titers.

### 2.4. DNA Extraction and PCR

DNA was extracted from blood (100 μL) and cheek swab samples using a DNA/RNA extraction kit (Litekh, Moscow, Russia). The *CCR5* genotype was determined by PCR using primers [13] spanning over the exon 4 region of the *CCR5* gene, which is known to be polymorphic. Amplification products were separated by gel electrophoresis. Functional and 32 nt deletion (Δ*32CCR5*) PCR products were visually discriminated (Figure 1).

### 2.5. Multiplex Analysis

Serum levels of 94 analytes was analyzed using Bio-Plex (Bio-Rad, Hercules, CA, USA) multiplex magnetic bead-based antibody detection kits following manufacturer’s instructions. Multiplex kits (Bio-Rad, Hercules, CA, USA), Bio Plex Pro Human Cytokine 21-plex, Bio Plex Human Cytokine 27-plex, Bio Plex Human Cytokine 37-plex, and Bio-Plex Human MMP Panel panels were used in the study. Data collected was analyzed using with MasterPlex CT control software and MasterPlex QT analysis software (MiraiBio, Alameda, CA, USA).

### 2.6. Statistical Analysis

Statistical analysis was conducted using R language for statistical computing [14], RStudio [15], and package “tableone” [16]. Illustrations were built with the “ggplot2” package [17]. Continuous variables were presented with their respective median (M), first (Q1), and third (Q3) quartiles. Categorical variables were presented with proportions and percentages. Comparisons were carried out using Mann–Whitney U tests for continuous and Fisher’s exact tests for categorical variables. The threshold used for statistical significance was *p* < 0.05.

Relevance of severity-associated continuous variables was reported as common language effect size (using package “canprot” [18]), rank-biserial correlation (the simple difference formula), as well as pseudomedian and its nonparametric 95% confidence interval (CI). In case of categorical variables, odds ratio (OR) and its 95% CI were calculated. In order to retain only the most prognostically valuable analytes, redundant features were removed using a pairwise spearman correlation analysis with packages “caret” and “reshape2” [19,20,21]. If two variables had correlation coefficients higher than 0.5, the mean absolute correlation of each variable was calculated. Then, the variable with the largest mean absolute correlation was removed.

Pathway analysis was done using the Pathway Studio MammalPlus (Elsevier, Amsterdam, The Netherlands). Analytes which differ significantly between groups were used for enrichment analysis. Only pathways with *p* < 0.05 after correction for the multiple comparison were selected. The Benjamini–Hochberg method was used to control the false discovery rate.

## 3. Results

### 3.1. CCR5 Genetics in NE Cases and Tatarstan Population

Distribution of functional type *CCR5* gene and the gene with 32-base-pair deletion (Δ*32CCR5*) was analyzed in 592 donors representing the general population of Tatarstan. Using PCR, the genotype was determined as functional *CCR5* homozygous, heterozygous, or Δ*32CCR5* homozygous. A total of 492 individuals (83.1%) was determined as having the functional *CCR5* homozygous genotype, while 94 (15.9%) individuals were heterozygous and only 6 (1.0%) individuals were Δ*32CCR5* homozygous.

In NE cases, the *CCR5* genotype distribution was similar to that in the general population, where 80 patients (81.7%) had functional *CCR5* homozygous genotype, 16 cases (16.3%) were heterozygous, and 2 (2.0%) were Δ*32CCR5* homozygous. Differences in *CCR5* genotype distribution between the general population and NE controls were not significant (Fisher’s exact test *p* = 0.516). Therefore, there is no evidence for *CCR5* genotype contributing to susceptibility to hantavirus infection.

### 3.2. Patient’s Clinical Characteristics

A total of 98 NE cases (80 males, 18 females) was recruited for this study. However, the low number of Δ*32CCR5* homozygous NE (total of 2) was not sufficient for statistical analysis; therefore, they were excluded. Hence, 96 NE cases were selected for the study, which will be referred to as homozygous (functional *CCR5*/functional *CCR5*) and heterozygous (functional *CCR5*/*Δ32CCR5*). Baseline characteristics were summarized in Appendix A. The median age of homozygous and heterozygous patients did not differ significantly (M = 37.00 years, Q1 = 29.00, Q3 = 51.00, *n* = 79 vs. M = 42.00 years, Q1 = 33.00, Q3 = 49.50, *n* = 15, respectively; Mann–Whitney U test *p* = 0.522). Most of the patients were male in both groups 64 (80.0%) males and 16 (20.0%) females for homozygous; 14 (87.5%) males and 2 (12.5%) females for heterozygous); there was no difference in the sex of patients between homozygous and heterozygous patients (Fisher’s exact test *p* = 0.728).

The duration of oliguria and thrombocyte counts differed significantly between homozygous and heterozygous NE cases. The oliguric phase was shorter in heterozygous when compared to homozygous NE cases (M = 0.00 days, Q1 = 0.00, Q3 = 0.00, *n* = 6 vs. M = 2.00 days, Q1 = 0.00, Q3 = 3.00, *n* = 49; Mann–Whitney U test *p* = 0.013, adjusted *p* = 0.106). This data suggests less impact on kidney function in heterozygous patients as compared to those with homozygous genotype. Additionally, the thrombocyte counts were significantly higher in heterozygous as compared to homozygous NE cases (M = 157.00 × 10^9^/L, Q1 = 86.00, Q3 = 241.00, *n* = 15 vs. M = 89.00 × 10^9^/L, Q1 = 64.00, Q3 = 125.00, *n* = 77; Mann–Whitney U test *p* = 0.018; adjusted *p* = 0.106). Also, the thrombocyte counts were significantly lower in homozygous NE as compared to controls (M = 214.00 × 10^9^/L, Q1 = 200.00, Q3 = 217.00, *n* = 9 vs. M = 89.00 × 10^9^/L, Q1 = 64.00, Q3 = 125.00, *n* = 77; Mann–Whitney U test adjusted *p* < 0.001). Other clinical parameters did not show significant differences between groups of patients. Thrombocyte counts are used to determine the risk for bleeding in hantavirus patients [22]. Our data suggests that heterozygous NE would have lower risk of developing hemorrhagic disturbances as compared to the homozygous genotype. Interestingly, there were two fatal NE case documented in the 2016 NE outbreak in the Republic of Tatarstan. PCR analysis of DNA extracted from the postmortem collected tissue revealed the homozygous functional *CCR5* genotype in both patients. Although these are two cases and more studies are required, it appears that the homozygous functional *CCR5* genotype is characterized in NE patients by more impact on kidney function and a higher chance of developing hemorrhages.

### 3.3. Assessment of Serum Analytes in NE Cases and Controls

A total of 94 analytes was measured including cytokines, chemokines, matrix metalloproteinases (MMPs), and soluble receptors in 80 homozygous NE cases, 16 heterozygous NE cases and 27 controls (median age 36.8 (Q1 = 23.5, Q3 = 49.3) years old). Eighty NE samples (median age 37.50 (range: 29.75, 51.50) years old) and twenty-seven controls (median age 36.8 (23.5, 49.3) years old).

The control *CCR5* genotype was analyzed before conducting this analysis to make sure that the genotype composition of the control represents that of the general population and resembles that of NE. Functional homozygous *CCR5* was 88.9%, while heterozygous *CCR5* was 11.1% in the controls. This distribution of *CCR5* genotype frequency in controls closely resembles that in the general population (83.1% and 15.9%, respectively) and in NE patients (81.7% and 16.3%, respectively).

The levels of 50 analytes were found to be differently expressed in homozygous NE cases as compared to control (Figure 2; Appendix A).

However, when adjusted *p*-values were calculated, the number of affected analytes was reduced to forty-six. These analytes included chemokines, growth factors, interferons, interleukins/receptor, MMPs, other cytokines, and tumor necrosis factors/receptor.

In the heterozygous NE cases, a lesser number of serum analytes (32 analytes) was found affected as compared to controls (Figure 3; Appendix A). After calculation of adjusted p-values, the level of 20 serum analytes remained significantly different from that in controls. These analytes included the following categories: chemokines, growth factors, interferons, interleukins, and MMPs. Cytokines and tumor necrosis factors/receptor were excluded in the heterozygous group when adjusted *p*-values were calculated. Also, the greater number of analytes was in the growth factor, IFN, IL, and MMP categories in homozygous as compared to heterozygous NE. Several cytokines such as sCD163, sIL6RA, PTX3, SPP1, CHI3L1, and thymic stromal lymphopoietin were affected exclusively in homozygous NE. These cytokines are involved in activation of inflammation, leukocyte migration, and differentiation [23,24]. Therefore, it could be suggested that homozygous NE will be characterized with more extensive inflammation, leukocyte migration, and differentiation as compared to the heterozygous one.

When 94 analytes were compared between homozygous and heterozygous NE, serum levels of nine molecules (MMP2, MMP3, MMP8, MMP9, MMP10, MMP12, MMP13, IL9, and IL32) differed significantly (Figure 4; Appendix A). Levels of IL9 were higher (*p* = 0.011) while levels of IL32 were lower (*p* = 0.014) in homozygous NE as compared to heterozygous cases (Figure 4; Appendix A). Interestingly, differences in serum levels of MMP2, MMP3, MMP8, MMP9, MMP10, MMP12, and MMP13 remained significant after adjustment for multiple comparisons. MMP serum levels were upregulated in homozygous cases as compared to that in heterozygous NE (Figure 4), suggesting that the matrix remodeling is more pronounced in homozygous NE.

### 3.4. Pathway Analysis in Functional CCR5 Homozygous and Heterozygous NE Cases

Only analytes which differ significantly between homozygous and heterozygous NE (MMP2, 3, 8, 9, 10, 12, and 13; IL9 and IL32) were included into the pathway analysis. Several pathways were identified as activated in NE cases, including inflammation and tissue damage (see Figure 5 for a typical pathway identified) and suggesting that extracellular matrix degradation and remodeling plays a significant role in functional *CCR5* homozygous NE pathogenesis. All these pathways have MMP activation in common, suggesting that extracellular matrix degradation and remodeling plays a significant role in functional *CCR5* homozygous NE pathogenesis.

### 3.5. Prognostic Value of CCR5 Genotype, PDGFBB, CCL27, and CXCL12

Duration of oliguria two days or longer and platelet count less than 150 × 10^9^/L were selected as criterion to define NE severity. Using these criterion, 31 patients had severe while 43 patients had mild forms of NE. Interestingly, 12 out of 43 mild NE were heterozygous, while only 1 out of 31 severe NE cases was heterozygous. These data support the association between homozygous *CCR5* genotype and the severe form of NE.

Next, these groups were compared to identify predictors of the disease severity. As potential predictors of the disease severity, 14 analytes were selected (Appendix A) in addition to CCR5 (*p* = 0.006, OR = 11.32, 95% CI: 1.50–511.31). These analytes include MMPs, cytokines, chemokines, and creatinine, which were shown to be involved in pathogenesis of hantavirus infection [25,26,27].

After exclusion of the redundant features (Appendix A), creatinine, MMP2, Tumor Necrosis Factor Ligand Superfamily Member 10 (TNFSF10), Colony Stimulating Factor 2 (CSF2), PDGFBB, IFNγ, IL32, CCL27, and CXCL12 were retained. When a random patient with severe NE was compared to a random patient with mild NE, the severe NE had higher serum levels of MMP2, creatinine, CSF2, CXCL12, and CCL27 than the mild form in 63.9%, 64.4%, 66.8%, 71.0%, and 71.0%, respectively, of such pairs. In contrast, higher levels of IL32, IFNγ, PDGFBB, and TNFSF10 were detected in only 24.6%, 28.5%, 31.7%, and 35.9% severe NE, respectively, while the mild form of the disease had higher rates of these cytokines upregulated (75.4%, 71.5%, 68.3%, and 64.1%, respectively) (Appendix A).

In line with that, MMP2, creatinine, CSF2, CXCL12, and CCL27 showed positive rank-biserial correlation with the severe disease, supporting the role of these cytokines in pathogenesis of this form of NE. There was negative rank-biserial correlation for TNFSF10, PDGFBB, IFNγ, and IL32, indicating that these cytokines could act to ameliorate severe symptoms of the disease (Figure 6).

In absolute terms, pseudomedians for comparison between mild and severe NE analyte levels were estimated as −113.19, −28, −3.39, −19.38, and −18.91 pg/mL for MMP2, creatinine, CSF2, CXCL12, and CCL27, correspondingly. For IL32, IFNγ, PDGFBB, and TNF-related apoptosis-inducing ligand (TRAIL) levels pseudomedian were 16.66, 33.74, 233.29, and 11.60 pg/mL, respectively (Appendix A).

After considering adjusted *p*-values, we conclude that PDGFBB, IFNγ, CCL27, and CXCL12 in addition to the *CCR5* genotype could be utilized as predictors of the severe form of NE.

## 4. Discussion

The key finding of our study is that the *CCR5* genotype affects the duration of the oliguric phase, thrombocyte counts, and serum MMP levels in NE cases. The shorter oliguric phase and higher thrombocyte counts found in heterozygous as compared to homozygous NE suggests amelioration of clinical symptoms in heterozygous NE patients and, as a result, a milder form of the disease. This is the first study where a genetic marker associated with the regulation of leukocyte trans-endothelial migration has been shown to affect the duration of the oliguric phase and thrombocyte counts, which are often used to determine the severity of the NE. The duration of the oliguric phase reflects the degree of kidney damage [28]. The low thrombocyte counts are an early sign of hemorrhagic disturbances, where the drop of thrombocyte counts below 50 cell/μL this indicates severe bleeding [29]. We have shown that the thrombocyte counts were higher in heterozygous NE, suggesting a lower likelihood of hemorrhagic complications. Our data suggests that NE patients with a heterozygous genotype will have a milder form of the disease and a lower likelihood of complications.

The cytokine storm hypothesis has been suggested to explain interrupted integrity of the endothelial barrier and leukocyte migration in HFRS. One of the most consistent findings in hantavirus cases is a high number of cytokine producing mononuclear leukocytes in the tissues [27]. The leukocyte migration is highly regulated process, involving CHI3L1, tymic stromal lymphopoietin, and sCD163. Several of these molecules are expressed by macrophages [23,24,30]. CCR5 is one of the main receptors involved in recruiting activated Th1 cells [7,31]. Also, CCR5 plays a crucial role in recruitment of memory CD8+ T cells into the tissue [32]. Interestingly, the phenotype of mononuclear leukocytes infiltrating tissues in hantavirus-infected patients closely resembles those expressing the CCR5 receptor [3,11]. Therefore, we suggest that expression of functional CCR5 receptor is essential for leukocyte trans-endothelial migration in NE.

The most striking observation was that serum MMP levels were significantly higher in homozygous as compared to heterozygous NE. MMPs are involved in breaking down the extracellular matrix (ECM) [33] and in releasing matrix-bound growth factors [34], which are essential for leukocyte migration. Th1 lymphocytes producing MMP2 and MMP9 have higher migratory capacities as compared to Th2 lymphocytes [35]. Also, homozygous NE patients had increased serum levels of MMP3 and MMP10, targeting fibronectin and laminin [36,37], as compared to heretozygous NE patients. Therefore, we suggest that MMP-induced ECM remodeling and leukocyte migration will be more evident in homozygous as compared to heterozygous NE patients.

Serum cytokine analysis revealed that heterozygous NE patients were characterized by less activation of the IL17 pathway as compared to homozygous cases. Interestingly, IL17 stimulates production of MMP3, 9, and 13 in monocytes and macrophages [38], while Th17 lymphocytes can produce MMP3 [39]. Coincidently, the levels of MMP3, 9, and 13 were higher in homozygous as compared to heterozygous NE. Therefore, we suggest that, in homozygous NE, tissue infiltrating leukocytes could contain Th17 populations. These leukocytes could locally produce cytokines stimulating and attracting neutrophils, thereby enhancing inflammation.

An analysis of the PDGFBB, IFNγ, CCL27, CXCL12, and CCR5 for prognostic value for diagnosis of severe NE revealed the predictive value of CCL27. We have demonstrated that increased CCL27 and being homozygous for the functional *CCR5* genotype are characteristic for severe NE. CCL27 is a chemoattractant for antigen-specific T lymphocytes [40]. Therefore, we suggest that antigen-specific T lymphocytes migrate more effectively across the endothelium in severe as compared to mild forms of NE.

## 5. Conclusions

In conclusion, our data suggest that the *CCR5* genotype could play a role in the pathogenesis of NE. Although the *CCR5* genotype does not contribute to the susceptibility to hantavirus infection, those patients homozygous for functional *CCR5* were characterized by a longer oliguric phase and lower thrombocyte counts when compared to heterozygous NE cases. Also, significant differences in serum levels of MMPs in homozygous and heterozygous NE cases were detected. Our data suggests differences in NE pathogenesis where homozygous cases have more pronounced disintegration of the ECM and increased activated Th1 lymphocyte migration across the endothelial barrier. Pathway analysis revealed activation of the Th17 pathway in homozygous NE cases. Our data, for the first time, presents the role of the *CCR5* receptor genotype in NE pathogenesis.

## Figures and Tables

**Figure 1 viruses-11-00675-f001:**
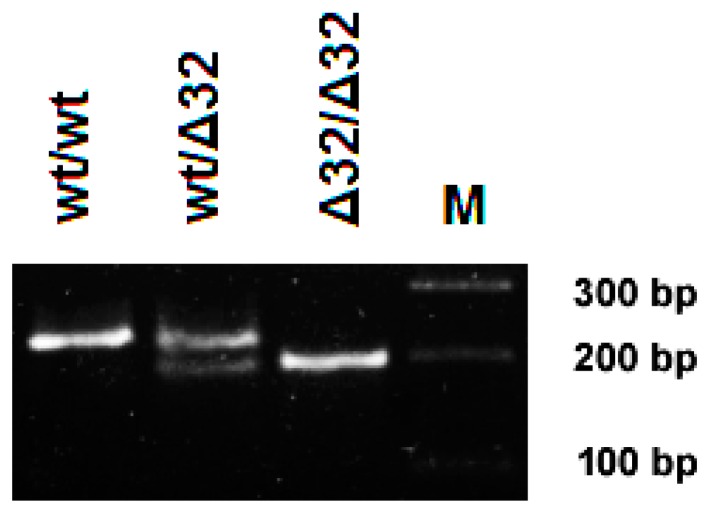
PCR analysis of *CCR5* genotype: Total DNA was extracted and used for PCR amplification. Expected sizes for functional *CCR5* are 225 bp and for Δ*32CCR5* are 193 bp. wt/wt—functional *CCR5* homozygous; wt/Δ32—functional heterozygous; Δ32/Δ32—non-functional Δ*32CCR5* homozygous; M—molecular weight marker.

**Figure 2 viruses-11-00675-f002:**
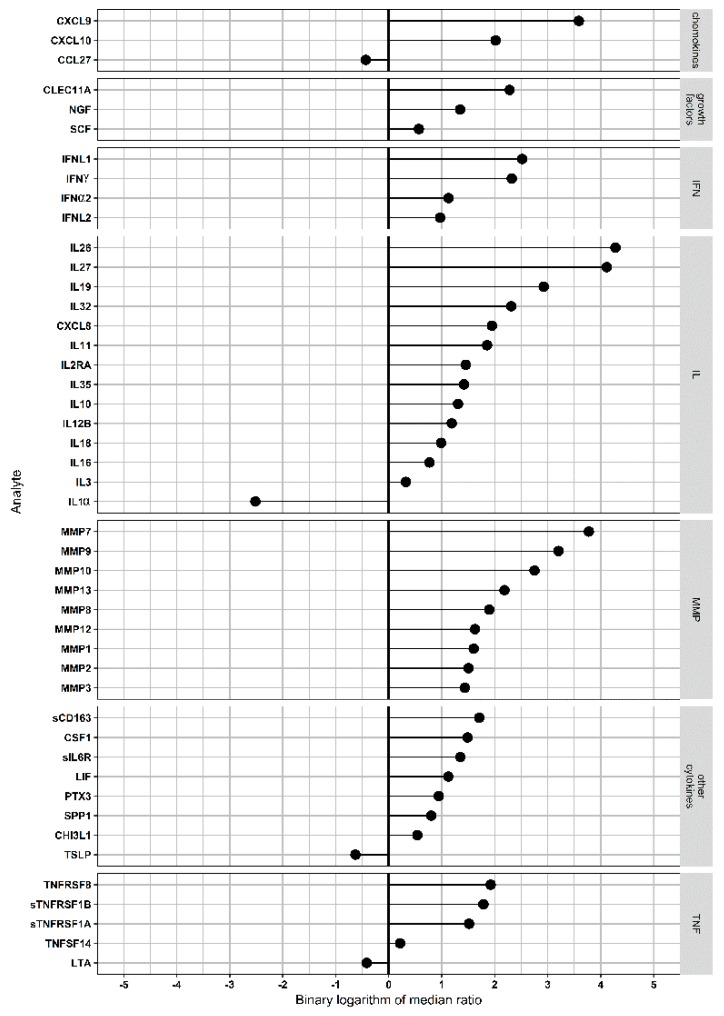
Analysis of functional *CCR5* homozygous serum multiplex data: Serum (50 μL) was used to detect serum levels of 94 analytes. All analytes were divided into TNF&Receptors, growth factors, IFN-related, IL-related, other cytokines, chemokine-related, and MMPs, and their levels were compared using the Mann–Whitney U test. The Benjamini–Hochberg method was used to control the false discovery rate, and only analytes with *p* < 0.05 were selected. For every analyte, the ratio of median level in functional *CCR5* homozygous patients to that in control was calculated. Then, binary logarithm of these values was used for sorting of analytes and generation of the “lollypop” diagram with the “ggplot2” package.

**Figure 3 viruses-11-00675-f003:**
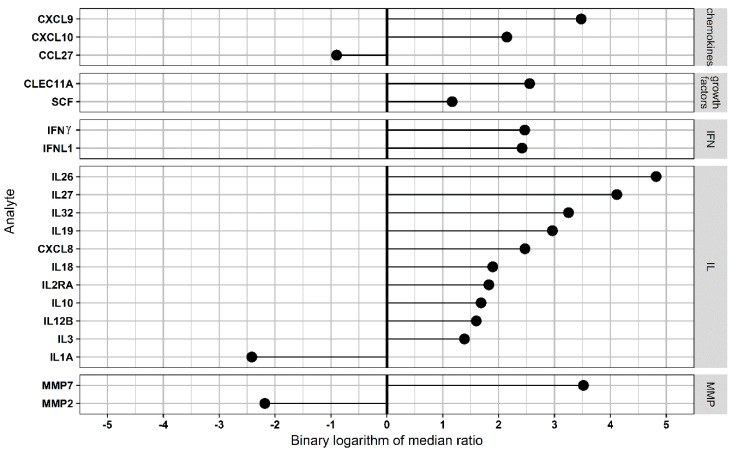
Analysis of functional *CCR5/Δ32CCR5* heterozygous serum multiplex data: Serum (50 μL) was used to detect serum levels of 94 analytes. All analytes were divided into TNF&Rs, growth factors, IFN-related, IL-related, other cytokines, chemokine-related, and MMPs, and their levels were compared using the Mann–Whitney U test. The Benjamini–Hochberg method was used to control the false discovery rate, and only analytes with *p* < 0.05 were selected. For every analyte, the ratio of median level in functional *CCR5*/Δ*32CCR5* heterozygous patients to that in control was calculated. Then, binary logarithm of these values was used for sorting of analytes and generation of the “lollypop” diagram with the “ggplot2” package.

**Figure 4 viruses-11-00675-f004:**
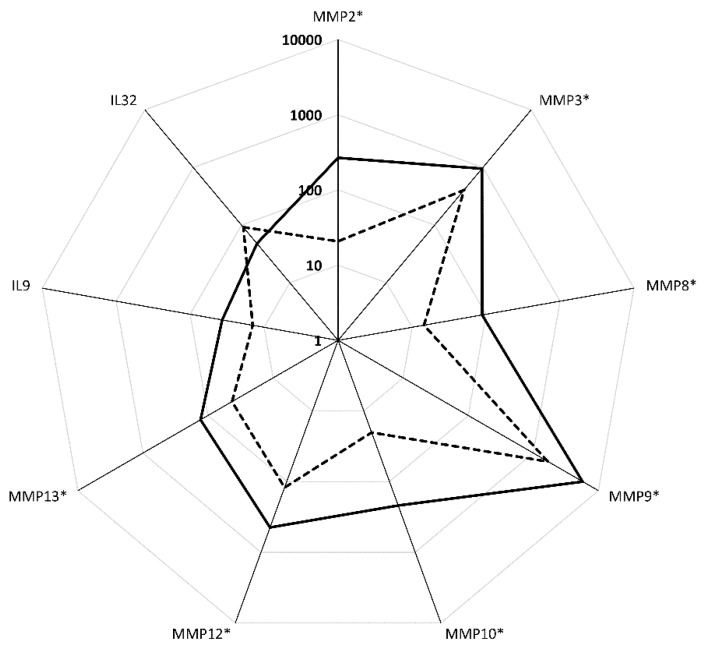
Analytes differed significantly between functional *CCR5* homozygous and functional *CCR5*/Δ*32CCR5* heterozygous NE cases. The Mann–Whitney U test was used to compare analyte levels between functional CCR5 homozygous and functional *CCR5*/Δ*32CCR5* heterozygous NE cases. Analytes with *p* < 0.05 were used to generate logarithmic scaled radar plot. The Benjamini–Hochberg method was used to control the false discovery rate. Asterisk marks those analytes differences that remained significant after adjustment for multiple comparisons. Solid line—functional *CCR5* homozygous patients; dotted line—functional *CCR5*/Δ*32CCR5* heterozygous patients.

**Figure 5 viruses-11-00675-f005:**
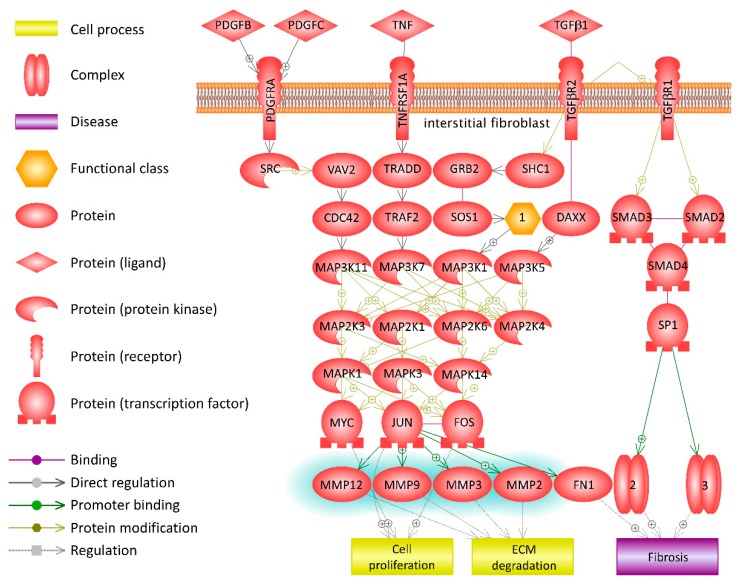
Pathway analysis of serum analytes in functional *CCR5* homozygous and functional *CCR5*/Δ*32CCR5* heterozygous NE—Interstitial Fibroblasts in Pyelonephritis. Pathway analysis was done using the Pathway Studio MammalPlus (Elsevier) tool. Analytes which differ significantly between groups were used for enrichment analysis. Only pathways with *p* < 0.05 after correction for the multiple comparisons were selected. Blue color highlights analytes which were significantly lower in functional *CCR5*/Δ*32CCR5* heterozygous NE.

**Figure 6 viruses-11-00675-f006:**
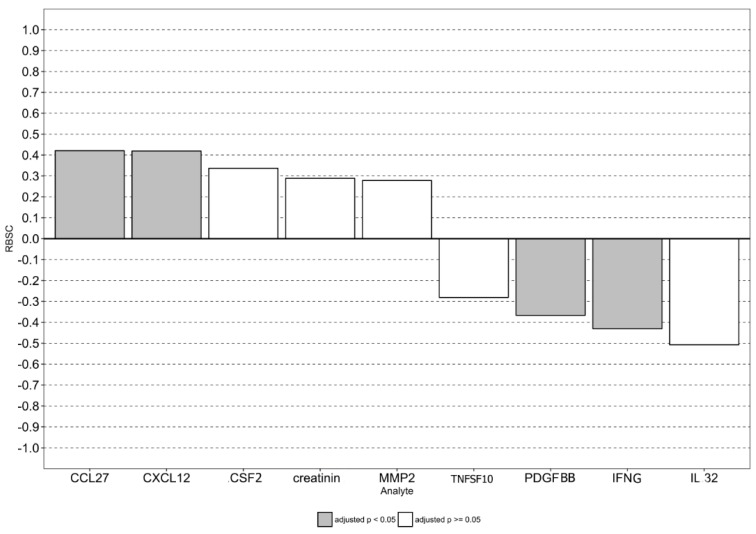
Rank-biserial correlation (the simple difference formula) analysis to assess the relevance of the severity-associated analytes: In order to retain only the most prognosis valuable analytes, redundant features were excluded using pairwise spearman correlation analysis with packages “caret” and “reshape2”. If two variables had absolute values of correlation coefficients higher than 0.5, the mean absolute correlation of each variable was calculated (only correlation coefficients significantly different from zero were considered). The variable with the largest mean absolute correlation was removed.

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
