# Peer review of "Δccr5 Genotype Is Associated with Mild Form of Nephropathia Epidemica"

_viruses, 2019, doi:10.3390/v11070675_

Round 1
Reviewer 1 Report
The manuscript by Kletenkov et al., describes the role of CCR5 genotypes in Puumala hantavirus-caused NE. By showing that CCR5 genotypes have a role in the severity but not the occurrence of the disease, this study reinforces the common idea in hantavirus field that the host immune response, but not the virus itself, play the most important part in disease pathogenesis. This is an important study and adds the accumulating evidence of host genetics affecting susceptibility to infectious diseases (and their pathogenesis). However, some points need to be addressed before publishing.
Major points:
1) The control groups used for clinical comparisons and multiplex analysis are poorly described. Age? What is the CCR5 genotype frequency in these groups? If the genotype frequencies have not been studied, then the control groups are not valid control group for either homozygous wtCCR5 or heterozygous CCR5 genotype NE groups. If no better description is provided, then one should stick to comparing only homozygous vs. heterozygous NE groups as one comparison and total NE cases to controls as the other.
Minor points:
1) The authors refer to full length, non-deleted, functional CCR5 gene as “wild type” but would it be more appropriate to talk about full-length or functional CCR5 in this case.
2) Why was the duration of oliguria and not serum creatinine used for describing severity of kidney dysfunction?
3) Rows 117-120: The method used to remove redundant features in correlation data is confusing in its present form. Needs some rewording to make it more clear.
4) Row 150: Why is the number of patient samples used for oliguria approximately half of that as for thrombocyte comparison? Would be good to clarify this discrepancy.
5) Row 239: What is the logic of choosing these analytes? Why were not all analyzed analytes used?
Author Response
Δccr5 Genotype Is Associated With Mild Form Of Nephropathia Epidemica
Reviewer 1
Open Review
(x) I would not like to sign my review report
( ) I would like to sign my review report
English language and style
( ) Extensive editing of English language and style required
( ) Moderate English changes required
(x) English language and style are fine/minor spell check required
( ) I don't feel qualified to judge about the English language and style
Yes | Can be improved | Must be improved | Not applicable | |
Does the introduction provide sufficient background and include all relevant references? | (x) | ( ) | ( ) | ( ) |
Is the research design appropriate? | (x) | ( ) | ( ) | ( ) |
Are the methods adequately described? | ( ) | ( ) | (x) | ( ) |
Are the results clearly presented? | (x) | ( ) | ( ) | ( ) |
Are the conclusions supported by the results? | (x) | ( ) | ( ) | ( ) |
Comments and Suggestions for Authors
The manuscript by Kletenkov et al., describes the role of CCR5 genotypes in Puumala hantavirus-caused NE. By showing that CCR5 genotypes have a role in the severity but not the occurrence of the disease, this study reinforces the common idea in hantavirus field that the host immune response, but not the virus itself, play the most important part in disease pathogenesis. This is an important study and adds the accumulating evidence of host genetics affecting susceptibility to infectious diseases (and their pathogenesis). However, some points need to be addressed before publishing.
Major points:
1) The control groups used for clinical comparisons and multiplex analysis are poorly described. Age?
The details of the control and NE groups were added into the text. The median age of HFRS patients was 37.50 [range: 29.75, 51.50]; while the age of controls was 36.8 [range 23.5,49.3].
What is the CCR5 genotype frequency in these groups?
If the genotype frequencies have not been studied, then the control groups are not valid control group for either homozygous wtCCR5 or heterozygous CCR5 genotype NE groups. If no better description is provided, then one should stick to comparing only homozygous vs. heterozygous NE groups as one comparison and total NE cases to controls as the other.
Functional homozygous CCR5 was 88.9%; heterozygous CCR5 was 11.1% in the control group. This distribution of CCR5 genotype frequency in the control group closely resembles that of the general population (83.1% and 15.9%, respectively) and NE patients (81.7% and 16.3%, respectively). These data are now incorporated into the manuscript (line 168).
Minor points:
1) The authors refer to full length, non-deleted, functional CCR5 gene as “wild type” but would it be more appropriate to talk about full-length or functional CCR5 in this case.
We would agree, “wild type’” has been replaced with “functional” throughout.
2) Why was the duration of oliguria and not serum creatinine used for describing severity of kidney dysfunction?
The oliguric phase of HFRS is considered the critical phase for the development of a fatal outcome with approximately 50% of the lethal HFRS cases appearing during this phase” (doi.org/10.1093/ndt/15.6.751).
3) Rows 117-120: The method used to remove redundant features in correlation data is confusing in its present form. Needs some rewording to make it more clear.
This text has been modified as suggested.
4) Row 150: Why is the number of patient samples used for oliguria approximately half of that as for thrombocyte comparison? Would be good to clarify this discrepancy.
Not all data was available for each patient.
5) Row 239: What is the logic of choosing these analytes? Why were not all analyzed analytes used?
These analytes were selected based on the fact that they have been linked with HFRS severity in previous studies and as representative of either kidney failure or inflammatory markers (from the major groups thought to be involved in HFRS pathogenesis
Creatinine: commonly used as a marker of kidney failure in clinical testing
MMP2, MMP10 : both selected as representative of metaloproteinases
TNF beta: pro-inflammatory marker
TRAIL: pro-inflammatory marker
GM.CSF: major stimulator of granulocytes and macrophages
HGF: acute phase cytokine (marker of injury and angiogenesis)
b.NGF pro-inflammatory marker
PDGF.bb: angiogenesis marker (released from platelets, marker of platelet activation)
IFN.g : major cytokine in antiviral respons
IL4: major Th2 (T cell) differentiation cytokine
IL32: pro-inflammatory cytokine
CCL27: cytokine important in lymphocyte homing
CXCL12 : chemokine important in leukocyte activation
Submission Date
24 May 2019
Date of this review
08 Jun 2019 09:32:09
Reviewer 2 Report
This manuscript examines the association between CCR5 genotype and the severity of hantavirus-related nephropathia epidemica in a cohort of patients from the Republic of Tatarstan. The authors suggest that CCR5 heterozygous patients have less severe disease than homozygous, and suggest several analytes that could have potential as prognostic markers. Unfortunately, my statistical knowledge is not sophisticated enough to understand all the tests the authors have applied to their data and it is not possible for me to comment on the validity of their conclusions. If correct, I think the authors’ findings are moderately interesting and add to our understanding of the clinical characteristics of hantavirus infection.
Minor points:
Introduction: It would be helpful to have more details on the role of CCR5 gene polymorphism in virus infection. The authors do cite a couple of papers, but I think a few sentences describing this would help put this study into context.
Line 76: Guessing this should be -80 degrees, not 80 degrees,
Author Response
Reviewer 2
Open Review
(x) I would not like to sign my review report
( ) I would like to sign my review report
English language and style
( ) Extensive editing of English language and style required
( ) Moderate English changes required
(x) English language and style are fine/minor spell check required
( ) I don't feel qualified to judge about the English language and style
Yes | Can be improved | Must be improved | Not applicable | |
Does the introduction provide sufficient background and include all relevant references? | ( ) | (x) | ( ) | ( ) |
Is the research design appropriate? | ( ) | ( ) | ( ) | (x) |
Are the methods adequately described? | ( ) | ( ) | ( ) | (x) |
Are the results clearly presented? | ( ) | (x) | ( ) | ( ) |
Are the conclusions supported by the results? | ( ) | ( ) | ( ) | (x) |
Comments and Suggestions for Authors
This manuscript examines the association between CCR5 genotype and the severity of hantavirus-related nephropathia epidemica in a cohort of patients from the Republic of Tatarstan. The authors suggest that CCR5 heterozygous patients have less severe disease than homozygous, and suggest several analytes that could have potential as prognostic markers. Unfortunately, my statistical knowledge is not sophisticated enough to understand all the tests the authors have applied to their data and it is not possible for me to comment on the validity of their conclusions. If correct, I think the authors’ findings are moderately interesting and add to our understanding of the clinical characteristics of hantavirus infection.
Minor points:
Introduction: It would be helpful to have more details on the role of CCR5 gene polymorphism in virus infection. The authors do cite a couple of papers, but I think a few sentences describing this would help put this study into context.
Line 76: Guessing this should be -80 degrees, not 80 degrees,
Agree: corrected.
Submission Date
24 May 2019
Date of this review
03 Jun 2019 19:22:27
Round 2
Reviewer 1 Report
All the points have been addressed